# Choice architecture interventions to improve diet and/or dietary behaviour by healthcare staff in high-income countries: a systematic review

Lena Al-Khudairy,[1] Olalekan A Uthman,[1] Rosemary Walmsley,[1,2] Samantha Johnson,[1] Oyinlola Oyebode[1]

¹Warwick Medical School, University of Warwick, Coventry, UK
²Worcester College, University of Oxford, Oxford, UK

**Correspondence to**
Dr Oyinlola Oyebode;
O.R.O.oyebode@warwick.ac.uk

## ABSTRACT

**Objectives** We were commissioned by the behavioural insights team at Public Health England to synthesise the evidence on choice architecture interventions to increase healthy purchasing and/or consumption of food and drink by National Health Service (NHS) staff.

**Data sources** MEDLINE, EMBASE, CINAHL, Cochrane Central register of Controlled Trials, PsycINFO, Applied Social Sciences Index and Abstracts and Web of Science were searched from inception until May 2017 and references were screened independently by two reviewers.

**Design** A systematic review that included randomised experimental or intervention studies, interrupted time series and controlled before and after studies.

**Participants** Healthcare staff of high-income countries.

**Intervention** Choice architecture interventions that aimed to improve dietary purchasing and/or consumption (outcomes) of staff.

**Appraisal and synthesis** Eligibility assessment, quality appraisal, data abstraction and analysis were completed by two reviewers. Quality appraisal of randomised trials was informed by the *Cochrane Handbook*, and the Risk of Bias Assessment Tool for Nonrandomized Studies was used for the remainder. Findings were narratively synthesised.

**Results** Eighteen studies met the inclusion criteria. Five studies included multiple workplaces (including healthcare settings), 13 were conducted in healthcare settings only. Interventions in 10 studies were choice architecture only and 8 studies involved a complex intervention with a choice architecture element. Interventions involving a proximity element (making behavioural options easier or harder to engage with) appear to be frequently effective at changing behaviour. One study presented an effective sizing intervention. Labelling alone was generally not effective at changing purchasing behaviour. Interventions including an availability element were generally reported to be successful at changing behaviour but no included study examined this element alone. There was no strong evidence for the effect of pricing on purchasing or dietary intake.

**Conclusion** Proximity, availability and sizing are choice architecture elements that are likely to be effective for NHS organisations.

**Trial registration number** CRD42017064872.

## Strengths and limitations of this study

► Examines interventions appropriate to improve diet in this important and unique population group: healthcare staff.
► Potential bias in identification of studies was minimised through a comprehensive search across major electronic databases relevant to the topic of this review, reference lists of relevant studies were screened and no language restriction was applied.
► To increase methodological rigour and minimise error and bias, each study included in this review was selected, assessed, data extracted and quality assessed by two review authors independently.
► Many included studies examined complex interventions, and it is not always possible to draw conclusions about the contribution of specific choice architecture elements to the outcomes.

## INTRODUCTION

The leading risk behaviour for disability and death in England, as in other high-income countries, is poor diet.[1] Worldwide, the proportion of adults who are overweight has increased to 36.9% of men and 38.0% in women.[2] Disease associated with poor diet costs the National Health Service (NHS) at least £5.8 billion per year.[3] Nearly two-thirds of adults in England are now overweight or obese.[4] The overall cost of obesity to wider society was projected to reach £27 billion in 2015.[5]

Healthcare settings have an important role to play in helping their staff tackle this issue. First, obesity is associated with absenteeism, presenteeism, early retirement, injuries, discrimination and litigation.[6–8] In England, NHS staff absence due to poor health costs at least £2.4 billion a year.[9] Improving employee health may improve productivity and performance of the workforce leading to improved patient care. Second, health professionals

play an important role in health promotion. However, a high proportion of healthcare staff do not engage in positive health behaviours themselves.[10 11] Nurses with poor health behaviour may be less likely to carry out health promotion activity.[12] A questionnaire survey of 540 pre-registered nurses found that three quarters felt that their physical appearance affected how they were perceived as a nurse.[12] Most respondents felt that patients would be more likely to heed their advice on health behaviours if they appeared to be following that advice themselves. 79.1% agreed that nurses should present themselves as role models for health behaviours.[12] Similarly a questionnaire completed by occupational health staff found that a fifth of them felt their own weight made it difficult to address the issue in patients.[13] By taking action to address diet and obesity for healthcare staff, these organisations and staff will be role models for others. While these are good reasons for any healthcare setting taking a role in supporting staff to eat healthily in the UK, an additional reason is that the NHS is the largest employer.[14] This means that improving the health and well-being of the 1.4 million NHS staff will lift the health and well-being levels of the entire UK population.

The WHO defined a healthy workplace as 'one in which workers and managers collaborate to use a continual improvement process to protect and promote the health, safety and well-being of all workers and the sustainability of the workplace'.[15] There are initiatives around the world to improve workplace well-being, and many of these are specific to healthcare settings.[16] In England, there are a number of policy drivers to encourage and support NHS organisations to improve the food and drink environment including an NHS Commissioning for Quality and Innovation Indicator: healthy food for NHS staff, visitors and patients.[17] Changing the environment to promote healthy behaviour, such as healthy food and drink purchasing and consumption, can be particularly useful and effective because

► When the environment makes healthy choices easier, this does not require any effort by the target audience. This means it is not just those people who are highly motivated to make healthy changes who benefit.
► Most changes are no-cost or low cost to introduce.

We were commissioned by the behavioural insights team at Public Health England (PHE) to undertake a systematic review and evidence synthesis, examining the literature to inform the evidence base on choice architecture interventions to increase healthier purchasing and/or consumption of food and drink by NHS staff.

## MATERIALS AND METHODS

The protocol for this systematic review was registered with PROSPERO (an international database of prospectively registered systematic reviews in health and social care) on 1 May 2017 (CRD42017064872).[18]

MEDLINE, EMBASE, CINAHL, Cochrane Central register of Controlled Trials, PsycINFO, Applied Social Sciences Index and Abstracts, and Web of Science were searched from inception until May 2017. Reference lists of retrieved included studies and of identified relevant systematic reviews were scanned for any additional studies not picked up by our electronic searches. No restrictions were made by language or year of publication. The search terms used included medical subject headings or the equivalent, and text word terms (ie, health promotion, choice architecture, food services, meals, workplace). A specialist librarian (SJ) was consulted for further search terms. Searches were tailored to individual databases (please see online supplementary material 1).

Studies met the inclusion criteria if they fulfilled all of the following:

Study design: Randomised experimental or intervention studies, including cluster randomised trials, quasi-randomised studies, interrupted time series studies (before and after) and controlled before and after studies.

Participants: NHS staff or staff from any other healthcare setting in a high-income country context. We used the World Bank classification for high-income countries, which includes 79 countries in total.[19]

Interventions: Behavioural interventions that aimed to change dietary purchasing and/or consumption of staff were included. We restricted the specific behavioural interventions examined to 'choice architecture' or 'nudge' or 'design' interventions. We used Hollands'[20] definition of choice architecture interventions—'those that involve altering small-scale physical and social environments, or micro-environments to cue healthier behaviour' and the associated framework to structure this review. We expanded this framework to include interventions that altered the price of items to cue healthier behaviour (table 1). This framework was chosen in discussion with PHE as it has previously been used to examine dietary interventions and was applicable to the context (healthcare settings).

Outcomes: -(a) dietary (food and/or drink) purchasing (eg, sales data; receipts analysis) in restaurants, canteens, vending machines or other situations in which food and drinks are sold; or (b) dietary intake assessed by validated tools or by more than one dietary measure such as multiple 24 hours dietary recalls. We did not examine caloric intake as self-reported measures showed poor correlation compared with objective measures for caloric intake.[21]

Following the database searching, titles, abstracts or both of every record retrieved were screened for potential relevance by two review authors (LA and OO) independently. Studies not carried out in high-income countries, qualitative studies and studies of children were excluded at this stage. Following this preliminary screening, full reports of potentially relevant studies were obtained, and two reviewers (two of LA, OO, OU) independently assessed studies for inclusion/exclusion using a checklist form based on the four inclusion criteria above. Where there

**Table 1** Typology of choice architecture intervention in micro-environments, modified from Hollands *et al* [20]

| Intervention class | Intervention type |
| --- | --- |
| Primarily alter properties of objects or stimuli | Ambience—alter aesthetic or atmospheric aspects of the surrounding environment |
| | Functional design—design or adapt equipment or function of the environment |
| | Labelling—apply labelling or endorsement information to product or at point-of-choice |
| | Presentation—alter sensory qualities or visual design of the product |
| | Sizing—change size or quantity of the product |
| | Pricing—change price of the product |
| Primarily alter placement of objects or stimuli | Availability—add behavioural options within a given microenvironment |
| | Proximity— make behavioural options easier or harder to engage with, requiring reduced or increased effort |
| Alter both properties and placement of objects or stimuli | Priming—place incidental cues in the environment to influence a non-conscious behavioural response |
| | Prompting— use non-personalised information to promote or raise awareness of a behaviour |

was disagreement about the inclusion of a study, a third reviewer was consulted (LA or OO or OU). Where resolution of a disagreement was not possible, we added the article to those 'awaiting assessment' and contacted study authors for clarification. Data were extracted from the included studies by two reviewers (two of LA, OO, OU, RW) independently using a predefined and pre-piloted data abstraction form (please see online supplementary material 2). Key data included details on methods, participants, intervention (including mapping to framework domains), outcomes, funding and notable conflicts of interest of study authors. The assessment of the methodological quality of included randomised trials was informed by guidelines from the *Cochrane Handbook*.[22] Risk of bias for other quantitative study designs used the Risk of Bias Assessment Tool for Nonrandomized Studies.[23]

### Patient and public involvement
As this was a themed commissioned piece of work, patient and public advisors were not involved in the development of the research question, design of the study or conduct of the study. We were required to develop a toolkit (online supplementary material 3) in a companion format designed for the audience who make decisions about catering in NHS organisations. Two advisors (please see acknowledgement) reviewed the toolkit to ensure its relevance to the target audience.

## RESULTS
Searching electronic databases generated 17 505 hits which equated to 14 294 individual records after duplicates were removed. Screening of titles and abstracts for potential relevance excluded 14 120 records as they did not meet the above-mentioned inclusion criteria of this review, leaving 174 potentially relevant records.

Screening relevant systematic reviews and included studies references yielded an additional 10 records. After formal inclusion/exclusion of 184 records, we identified 28 records from 18 studies met the inclusion criteria of this review. This included nine records from five studies in which the settings were not exclusively hospitals or other healthcare workplaces, but in which at least one of the sites involved was relevant to our review. In addition, we noted 10 records from 9 studies where we were unable to ascertain whether the study met our inclusion criteria based on information included in the study reports, and author contacts received no replies. In these cases, studies were classified as 'awaiting classification' (figure 1).

Results were narratively synthesised due to the substantial heterogeneity in study design, interventions (even when classified within the same framework category) and outcomes reported. We were unable to proceed with planned meta-analysis or subgroup analyses.[18]

### Included studies
#### Healthcare settings
Key characteristics of 13 included published studies are presented in table 2. Eight studies[24–33] examined pure choice architecture interventions and five studies[34–41] examined complex interventions for which some elements were choice architecture. One included study was conducted in the UK,[24] three in other European countries,[25 26 31] eight in the USA[27–30 32–37 39–41] and one in Australia.[38] Eight studies were interrupted time series,[25 27–34 38] two were cluster randomised controlled trials (RCTs),[35 36 39–41] one was an RCT,[37] one was a controlled before and after study[26] and one cross-sectional study.[24] Seven studies examined purchasing outcomes,[25 27–31 38] five examined dietary

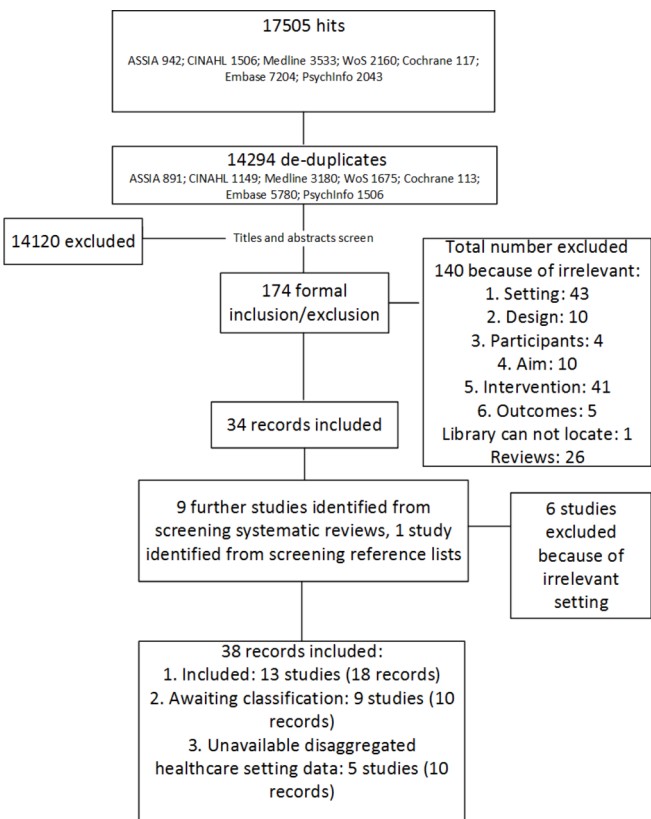

**Figure 1** Preferred Reporting Items for Systematic Reviews and Meta-Analyses flow diagram.

consumption[24 26 35–37 39–41] and one study examined both purchasing and dietary consumption.[34]

### Multiple workplaces

Five studies (table 2) in which settings included multiple workplaces and data were not disaggregated for healthcare settings were included in this review. Two studies[42–45] examined pure choice architecture interventions and three studies[46–49] examined complex interventions for which some elements were choice architecture. Two studies were conducted in the USA,[46 47] two in The Netherlands[42 45 48] and one in the UK.[43 44] Three studies were cluster RCTs,[42 45–47 49] one was a controlled before and after study[48] and one was a cross-sectional study.[43 44] Four studies examined dietary consumption[43 44 46–48] while one examined purchasing.[42 45] Further details of included studies are available in table 2.

### Awaiting classification

Despite contacting authors we were unable to classify nine studies because sufficient details were lacking in study reports. Five studies were unclear on the setting[50–54]; two studies were unclear on the outcomes[55 56]; one study was unclear on the type of participants[57] and one abstract was unclear on several domains.[58]

### Intervention components and effectiveness

We examined effectiveness of interventions in included studies by whether they were a choice architecture-only intervention or a complex intervention with choice architecture elements. Summaries of the components included in interventions mapped against their reported effectiveness are presented in tables 3 and 4.

### Pure choice architecture

Of the choice architecture-only interventions, almost all included studies involved labelling elements, including three studies that examined labelling in isolation. The most frequently studied single-element choice architecture interventions were proximity interventions, investigated in five studies.[24 25 27 30 31] No included study investigated ambience or priming elements (table 3). Most of the identified published studies reported successful interventions.

There was mixed evidence for the effectiveness of labelling interventions implemented on their own. Labelling interventions were not found to have an effect on purchasing in two studies,[25 29] although they were reported to change purchasing behaviour in one other study.[28] In the study that reported successfully changing purchasing through labelling, every item of food on offer in a hospital canteen was labelled with its caloric value. This had a significant effect on the number of calories purchased during the intervention phase. This study was at low risk of bias in three domains; however, it is more than four decades old and therefore likely to be examining a slightly different context to present day.[28] There was a low risk of bias in one of the studies that found no effect of labelling.[25] In this study, lean (30% less fat) and butter croissants which were similar in price, appearance and taste were on offer in a Dutch hospital restaurant, where they are a popular choice for lunch. During the intervention period, the lean basket was labelled with a sign that stated 'The saturated (unhealthy) fat content of this croissant is 30% lower'; however, there was statistically significant difference between the number or ratio of the two alternative products purchased during the intervention period compared with the control periods.[25] Another study labelled the two entrees on offer in a hospital canteen (one of which was designed to be healthier) with information on their constituent calories, fat, and sodium but this did not significantly change sales of the healthier option.[29] This study had a high risk of bias from confounding.[29] Overall it appears that labelling is unlikely to be effective on its own.

One study, with a low risk of bias in three domains, examined an intervention which used sizing alone, specifically, offering a smaller size of the main meal of the day, which was successful in terms of changing purchasing behaviour— about 10% of hot meals sold were small meals, with no overall change in total hot meals sold and no compensatory purchasing, for instance, purchasing the smaller hot meal and adding a portion of chips.[42 45] A pricing element was also evaluated in this study, in which the smaller meal was either priced proportionately so that both size and price of the meal were 65% of the standard; or 'value' pricing was used in which the size of the

**Table 2** Characteristics of included studies

| Study | Participants/worksite | Interventions | Outcomes |
|---|---|---|---|
| Dorresteijn et al[25] | Canteen, medical centre. University Medical Centre with >11 000 employees, >1000 hospital beds and >2000 customers visiting the hospital restaurant each day. *Inclusion/exclusion*: N/A (data not collected from individuals). | (1) Point-of-decision prompts on hospital elevator doors promoting stair use. In the hospital restaurant: (2) point of- purchase prompts promoting reduced-salt soup. (3) Point-of-purchase prompts promoting lean croissants. (4) Reversal of the accessibility and availability of diet margarine and butter. *Comparator*: No comparator. | Number and ratio of purchased normal-salt soup, reduced-salt soup, croissants, lean croissants purchased, diet margarine and butter. |
| Geaney et al[67] | Canteen, hospital. Participants were representative sample of 100 adults aged 18–64 years working in two public sector hospitals (50 staff from each hospital). *Inclusion*: Being an employee and consuming at least one main meal in the hospital staff canteen daily. *Exclusion*: None reported. | Restricting food high in salt, fat and sugar. Modifying menus to make healthier options available. Replacing purchasing orders for high salt products (eg, bacon) with low salt alternatives (eg, turkey). Salt removed in all cooking places and tables but available in small sachets at service. Nutrition information on salt reduction and a healthy diet displayed. No sauces or accompaniments added without customer's consent. Staff encouraged to consume extra salad and vegetable options at no extra charge. Cooking oil use limited. All desserts fruit based. *Comparator*: Canteen at another hospital with no intervention. | Macronutrients (g/day): total sugars, sucrose, fructose, total fat, saturated fat, protein, carbohydrates, salt. Micronutrients (mg/day): K, Ca, Fe, vitamin B6, vitamin B12, vitamin C, vitamin D. |
| Lassen et al[26] | Canteen, hospital. The canteen was seeking Keyhole certification in Denmark. Participants were canteen customers approached at random after purchasing lunch. *Inclusion*: Purchasing lunch in the canteen. *Exclusion*: Not reported. | Introduction of the Keyhole symbol on freshly prepared food in the canteen. The Keyhole symbol is used in the Nordic countries as a sign of a healthy choice (meeting certain criteria). *Comparator*: No intervention. | Energy per meal consumed (kJ), energy density of meal (kJ/100 g), fat content (E%), fruit and vegetables (g/100 g), salt (g/100 g), refined sugars (g/100 g), wholegrain (g/100 g). |
| Lowe et al[34] | Canteens, hospitals. *Inclusion*: Reporting eating lunch in the hospital cafeteria at least two times each week, on average, being between 21 and 65 years of age and being a hospital or university employee. *Exclusion*: Having a current diagnosis of a chronic disease or condition known to affect appetite or bodyweight, taking medication known to affect appetite or bodyweight, being pregnant or planning to become pregnant within the next 24 months, being enrolled or having plans to label within the next 24 months in an organised weight management programme, and/or having plans to terminate hospital employment within the next 12 months. | When the intervention period began, participants in both groups were exposed to environmental change (EC): reductions in the energy density (ED) of some foods offered in the cafeteria and introduction of nutritional labels for all foods sold in the cafeterias. Participants in the EC-Plus condition received two additional intervention components: training in reducing the ED of their diet and discounts on low-ED foods purchased in the cafeteria. *Comparator*: Environmental change only. | Fruit (servings/day), vegetables (servings/day), bread products (servings/day), dairy products (servings/day), fats and sweets (servings/day), meats (servings/day); Purchased kcal (kcal): purchased calories from fat (%); purchased calories from protein (%); purchased calories from carbohydrate (%). |
| MacDonald et al[38] | Aboriginal Community controlled health organisations. *Inclusion/exclusion:* Not reported. | A tailored 'Healthy Catering Toolkits' to local caterers at each site and included order forms classified according to the 'traffic light guide' to ensure healthy catering choices were easy ordering choices for staff. They also distributed 'traffic light guide' posters and information sheets, a nutrition policy template and a wallet-sized card for interpreting food labels. Comparator: No comparator. | Proportion of purchased foods in category 'foods to limit' (%). |
| Meyers[27] | Canteen, hospital. A hospital cafeteria that served hospital staff, students and hospital visitors. *Inclusion/exclusion*: Not reported. | Desserts were arrayed in columns four deep along the cafeteria counter. On control days research assistants arrayed desserts in columns of four, high-calorie desserts alternating with low-calorie desserts, permitting equal access to each dessert. High-calorie desserts were placed in the two front positions with low-calorie desserts in the read (less accessible) positions on days 3 and 5. On days 2 and 4 this order was reversed. *Comparator*: No comparator. | Dessert purchases. |
| Milich[28] | Canteen, hospital. *Inclusion*: Being a female employee (identifiable by an ID badge and/or uniform) and having a food tray. *Exclusion*: Being a patient or visitor, being visibly pregnant or being an employee who had brought food from home or only bought part of her meal in the cafeteria. | Calorie signs were posted for foods in the cafeteria. The intervention consisted of 1 week of price increase, followed by 1 week of price increase and calorie presentation. *Comparator*: No comparator. | Calories bought (kcal), money spent ($). |
| Racette et al[35] | Worksites, medical centre. *Inclusion*: All employees >18 years of age were eligible, including those who smoked, had pre-existing disease (eg, hypertension, diabetes) or used medications. *Exclusion*: Not reported. | The intervention comprised a constellation of nutrition components, physical activity components, and incentives designed to promote healthy dietary and physical activity behaviours, with the goals of promoting weight control and reducing risk factors for cardiovascular disease. Specific intervention components included weekly healthy snack cart, monthly lunchtime seminars, monthly newsletters, walking maps, participation cards and participation rewards. Each week a registered dietitian/exercise specialist was available at the worksite to provide support. *Comparator*: Health assessments only. | Fruit and vegetable intake (servings/day). |

**Table 2** Continued

| Study | Participants/worksite | Interventions | Outcomes |
|---|---|---|---|
| Sato et al[29] | Cafeteria, medical centre. Customers of the medical centre cafeteria aged 18 years and older. Roughly 70% of the customers at Kaiser Permanente San Francisco Medical Center are Kaiser employees; the rest are visitors or patients in the hospital. The cafeteria serves roughly 100 customers per day. *Inclusion/exclusion*: N/A (receipts collected anonymously). | A 'Healthy Pick' option meeting various criteria was made available every day. The meals available were labelled with information featuring calories, fat and sodium. *Comparator*: No comparator. | Number of Healthy Picks purchased, number of main meals purchased. |
| Sorensen et al[36 39–41] | Community health centres. *Inclusion*: Being a permanent employee and working at least 15 hours a week. *Exclusion*: Not reported. | The core intervention included periodic exposure to national 5-a-Day campaigns and a general nutrition presentation. In the additional worksite intervention, educational activities and changes were made to the worksite environment including labelling and adding fruit and vegetables to vending machines. *Comparator*: Core intervention only. | Fruit and vegetable intake (servings/day). |
| Stites et al[37] | Canteen, hospital. *Inclusion*: Having a body mass index of at least 25.0 kg/m², eating at least three lunches/week in the hospital cafeteria, being willing to allow researchers to collect data about their lunch purchases, and having access to a computer at work. *Exclusion*: Having a diagnosis of unstable hypertension, dyslipidaemia or coronary artery disease whose medical therapy had changed in the past three months, having plans to terminate hospital employment within 4 months following study enrolment, or being pregnant. | During the Full Intervention phase participants received mindful eating training, were encouraged to pre-order their lunches, and were given vouchers to use on lunch purchases. Following this there was a Partial Intervention phase, where participants were encouraged to preorder their lunches but did not receive vouchers. The preordering system listed the food available daily in the cafeteria, along with nutritional information. Employees had to order at least 45 min ahead of scheduled pick up time. *Comparator*: Wait list control. | Kcal per lunch purchased (kcal), fat grams per lunch purchased (g). |
| Thorndike et al[30 32 33] | Canteen, hospital. Hospital employees have the option of paying for cafeteria purchases by direct payroll deduction using a 'platinum plate' card. *Inclusion*: Using 'platinum plate' and making a purchase in the cafeteria at least three times during each 3-month period from baseline to the end of follow-up. *Exclusion*: Not reported. | A traffic light food labelling system was introduced, along with signage explaining it. Three months later further changes were introduced, including rearranging items in the beverage and sandwich refrigerators to put all the green items at eye level; placing baskets of bottled water throughout the cafeteria; and providing pre-packaged salads next to the pizza counter. *Comparator*: No comparator. | Of items bought: % red, % green (red labels and green labels as denoting unhealthy and healthy choices respectively). Of beverages bought: % red, % green. |
| Van Kleef[31] | Canteen, hospital. *Inclusion/exclusion*: N/A. | Four successive weeks were randomly assigned to four experimental conditions: Intervention 1: 25% of products healthy, these located at top shelves. Intervention 2: 25% of products healthy, these located at bottom shelves. Intervention 3: 75% of products healthy, these located at bottom shelves. Intervention 4: 75% of products healthy, these located top at shelves. *Comparator*: No comparator. | Number of healthy snacks sold, number of unhealthy snacks sold. |
| Beresford et al[46] | 28 worksites (six health service organisations). Inclusion: (for worksites) Having a food-serving cafeteria and having between 250 and 2000 employees. *Exclusion*: N/A. | The intervention was tailored to worksites. Elements included a kick-off event, the 5-a-Day message being posted on boards in each worksite, more fruit and vegetables becoming part of the menus and the provision of a self-help manual for every employee. *Comparator*: No intervention. | Fruit and vegetable intake (servings/day). |
| Holdsworth[43 44] | Six worksites (two healthcare). *Inclusion/exclusion*: Not reported. | The Heartbeat Award scheme increases opportunities for behaviour change by providing customers with information, reminders and reinforcement to guide them towards healthier choices. *Comparator*: No intervention. | Daily intake of biscuits and cakes, sweet puddings, confectionery, sugary drinks, processed meat, hard cheese, crisps, low-fat cheese, fried food, beans and pulses, fruit, vegetables, chicken and fish, starchy foods, red meat, low calorie drinks. All reported (1) no change/negative change and (2) positive change. |
| Kwak et al[48 59] | 12 worksites (two hospitals). *Inclusion*: Employees with body mass index >18 kg/m² and not under any medical restrictions. *Exclusion*: Not reported. | There was an individual and an environmental component to the intervention. Interventions included changes in the assortment of food products in the cafeteria, workshops, an information wall containing information on the balance between food intake and physical activity, posters or prompts stimulating stair use, and ways to form lunch-walking and cycling groups. | Intake of fibre-rich bread, low-fibre bread, fibre-rich main meal, low-energy-dense toppings, high-energy-dense toppings, low-energy-dense main meal, high-energy-dense main meal, low-energy-dense snacks, high-energy-dense snacks, low-energy-dense drinks, high-energy-dense drinks (all in servings/day). |
| Sorensen et al[47 49] | 16 worksites (at least one intervention and one control site specialising in 'healthcare'). *Inclusion*: (for worksites) Having 200–2000 employees, having a cafeteria with a kitchen, having an annual turnover rate of <25%, having <25% of employees working rotating shifts, part time or off site and being stable as a company, defined as having no plans for geographic relocation or major lay-offs in the next 2 years. *Exclusion*: Not reported. | Worksites received a 15-month intervention with various elements. These included direct education programmes, such as classes offered in all intervention worksites and optional activities tailored to individual worksites, and environmental programmes which targeted cafeterias both to increase the availability of low-fat, high-fibre foods and to provide supportive nutrition education by labelling those food options. *Comparator*: No intervention. | Total dietary fat (% kcal), total dietary fibre (g). |

Continued

**Table 2** Continued

| Study | Participants/worksite | Interventions | Outcomes |
|---|---|---|---|
| Vermeer et al[42 45] | 25 worksites (15 hospitals). *Inclusion*: (for worksites) Selling on average 30 meals/day, offering a reducible meal at least twice per week, being willing to be randomly allocated to a study condition, following the study protocol and providing their daily sales figures of hot meals and fried snacks. *Exclusion*: Not reported. | Intervention 1: A smaller portion (ie, about two-thirds of the size of the existing portion) was offered in addition to the existing portion and proportional pricing was employed (ie, the price was 65% of the existing size). Intervention 2: A smaller portion was added to the assortment and value size pricing (ie, a lower price per unit for large portions than for small portions) was employed (ie, the price was 80% of the existing size). *Comparator*: No intervention. | Number of large meals sold, number of small meals sold, number of fried snacks sold. |

N/A, not applicable.

meal was 65% of the standard, but the price only reduced to 80% of the standard. The pricing condition did not change the effectiveness of the intervention; there was no significant different between the proportionate and value pricing conditions.[42 45]

Proximity interventions alone were successful at changing purchasing behaviour in two studies.[25 27] The interventions included having margarine available in four centrally located and easy-to-reach positions in the hospital restaurant, but butter stored in a fridge, therefore requiring more effort to obtain which reduced sales of butter and increase sales of margarine[25]; rearrangement of desserts in a hospital canteen so that low-calorie desserts (fruit and jelly) or high-calorie desserts (pies and cakes) were at the front in most accessible positions.[27]

Other successful pure choice architecture interventions including more than one element were

▶ Labelling + availability interventions which changed dietary consumption in two studies.[26 43 44] Both these studies involved stipulating the proportion of meals on offer in a canteen that should be healthy (half in Lassen *et al*; a third in Holdsworth *et al*) and labelling these healthy meals as such. These are the only two pure choice architecture interventions which assessed dietary consumption. However, both had a high risk of bias over more than one domain.

▶ Labelling + proximity + prompting which changed purchasing behaviour in one study with a low risk of bias over most domains.[30] In this study, all items on offer were labelled according to a traffic light system, green items were put at eye-level while red items were made less visible, and permanent signage and menu board changes were made to highlight healthy options.

▶ Availability + proximity intervention which changed purchasing behaviour in one study with a low risk of bias.[31] In this study, a set of shelves stocked with snacks was placed on the check-out counter of a hospital canteen. In two phases, the shelves were stacked with 75% healthy and 25% unhealthy snacks and for the other two phases these proportions were reversed. In addition, the shelves were rearranged so that the healthy snacks were at the top in two phases, and at the bottom (requiring stooping to access) in the other two phases. Both availability (proportion of healthy snacks on offer) and proximity (ease of obtaining the healthy snacks) had a positive effect on healthy snack purchasing.

Finally, Geaney *et al*[24] used an intervention with several choice architecture elements which was successful in changing dietary consumption. In this study, menus were modified to increase availability of healthy options: high-salt products and processed meat were replaced with low-salt options, all desserts were fruit based, cooking methods with oil were limited. In addition, salt was removed in all cooking processes and from tables in the staff canteen (although it was available in small sachets from service). Nutritional information on salt reduction and a healthy diet was displayed in the canteen area. No sauces or accompaniments were added to any meals without the customer's consent, and extra salad and vegetable option was provided at no extra cost. This study had a low risk of bias and demonstrated an effect on intake of total sugars, total fats, saturated fats and salt as measured by 24-hour dietary recall data.[24]

The only unsuccessful multicomponent choice architecture-only intervention was examined in Dorresteijn *et al*[25] which used labelling + proximity + prompting. In this case, the two soups usually on offer in the restaurant were altered so that one option every day was reduced salt. Signs advertised that this soup 'contains 30% less salt and contributes to a healthy blood pressure'; however, this did not affect the number or ratio of normal salt to reduced-salt soup cups sold.

## Complex interventions including choice architecture

As with the choice architecture-only interventions, most of the complex interventions included labelling elements in studied interventions while none included ambience or priming elements. In addition, none of the included studies of complex interventions examined presentation or sizing interventions. Where choice architecture interventions were implemented as part of complex interventions (usually involving other elements of behaviour change, eg, educational elements), there were many examples of effective interventions (table 4).

Complex interventions involving an availability choice architecture element were effective at changing dietary consumption in two studies.[35 46] In one of these studies, the other elements of the intervention were extensive

**Table 3** Choice architecture elements and effectiveness of interventions examined in included studies

| Intervention class | Intervention type | Dorrestejin | | | Geaney | Holdsworth [43] | Lassen | Meyers [27] | Milich [28] | Sato et al [29] | Thorndike et al [30] | Van Kleef [31] | Vermeer et al [42] | |
|---|---|---|---|---|---|---|---|---|---|---|---|---|---|---|
| Studies | | a | b | c | | | | | | | | | a | b |
| Primarily alter properties of objects or stimuli | Ambience | | | | | | | | | | | | | |
| | Functional design | | | | X | | | | | | | | | |
| | Labelling | X | X | | X | X | X | | X | X | X | | | |
| | Presentation | | | | X | | | | | | | | | |
| | Sizing | | | | | | | | | | | | X | X |
| | Pricing | | | | | | | | | | | | | |
| Primarily alter placement of objects or stimuli | Availability | X | | | X | X | X | X | | | X | X | | |
| | Proximity | | | X | X | | | | | | X | X | | |
| Alter both properties and placement of objects and stimuli | Priming | | | | | | | | | | | | | |
| | Prompting | X | | | | | | | | | X | | | |
| Outcome | Effect on purchasing? | N | N | Y | | | | Y | Y | N | Y | Y | Y | Y* |
| Outcome | Effect on dietary consumption? | | | | Y | Y | Y | | | | | | | |

X blue box indicates element was implemented in the study.

*Although sizing plus pricing had an effect compared with control, there was no significant effect between sizing and pricing and pricing and sizing alone in this study.

N, not effective; Y, effective.

**Table 4** Choice architecture elements included in complex interventions and their reported effectiveness in included studies

| Intervention class | Intervention type | Berefsord | Kwak et al [48] | Lowe et al [34] | Macdonald et al [38] | Racette et al [35] | Sorensen et al [47] | Sorensen et al [36] | Stites et al [37] |
|---|---|---|---|---|---|---|---|---|---|
| Primarily alter properties of objects or stimuli | Ambience | | | | | | | | |
| | Functional design | | | | | | | | X |
| | Labelling | | | X | X | | X | X | X |
| | Presentation | | | | | | | | |
| | Sizing | | | | | | | | |
| | Pricing | | | X | | | | | |
| Primarily alter placement of objects or stimuli | Availability | X | X* | X | | X | X | | |
| | Proximity | | | | | | | X | X |
| Alter both properties and placement of objects and stimuli | Priming | | | | | | | | |
| | Prompting | | X* | | X | | X | | |
| Outcome | Effect on purchasing? | | | | Y | | | | Y |
| | Effect on dietary consumption? | Y | Y | Y | | Y | Y | N | |

X blue box indicates element was implemented in the study.
*Kwak et al used an assets-based approach in which intervention worksites chose intervention elements that suited their context. Examples of actions given were classified as 'availability' and 'prompting'.
N, not effective; Y, effective.

including pedometers, on-site WeightWatchers group meetings, an on-site group exercise programme, monthly lunchtime seminars, monthly newsletters, walking maps, team competitions, participation cards and participation rewards and weekly access to a registered dietitian/exercise specialist.[35] The environmental component of the intervention was a weekly healthy snack cart that visited staff.[35] In the other study, the intervention included formation of employee advisory boards in each worksite to implement activities in their workplace including education and information about 5-a-Day, and a self-help manual provided to every employee to support them making healthy choices. In addition, the 'availability' component was providing more fruits and vegetables as part of the regular menus in the worksite canteens.[46] In both cases, it is difficult to draw much of a conclusion about the 'availability' components compared with the other components of the intervention.

Labelling + prompting was a successful combination when used alongside a complex intervention in one study[38] and labelling + prompting + proximity was effective as part of a complex intervention in Sorensen et al[47, 47 49]. However, labelling + proximity, without prompting, was not effective in another study.[36 39–41] In all three studies, the other (non-choice architecture) elements of the interventions were direct educational activities, but none were particularly intensive suggesting labelling + prompting might work as pure choice architecture (labelling + prompting

+ proximity was also effective in the choice architecture-only studies).[30]

Labelling + pricing + availability was effective when used alongside a complex intervention in one study.[34] In this study, canteens modified the recipes of some foods offered in order to reduce the energy density and introduced nutritional labels for all foods. In addition, some employees were given training in reducing the energy density of their diet and discounts on lower-energy-dense foods bought in the canteen.[34] This study had high risk of attrition bias and possible evidence of selective outcome reporting. The study reported a significant reduction in total kilocalories purchased in the canteen and in the proportion of energy from fat.[34]

Stites et al[37] used a complex intervention with many choice architecture elements and was effective at changing behaviour. In this intervention, the participants received 'mindful eating training' and the hospital implemented an online preordering system for staff lunch. The preordering system had nutritional information displayed by all the choices, and for 4 of the 8-week intervention period, the employees received a discount for preordering rather than buying lunch direct in the canteen. Other elements included a preordering system which defaulted to the healthiest choice (requiring additional effort to select less healthy options). This study had low risk of bias and was effective at improving nutritional content of purchased lunches.

 

In one study, in which the intervention had an effect on dietary consumption, choice architecture elements were chosen and implemented according to what suited the local context with examples relating to availability (which also changed behaviour in two other studies) and prompting (which is not studied in isolation in any other study).[59]

## Methodological quality of included studies

The risk of bias of included studies is presented in tables 5 and 6. High risk of bias was seen in two of the six RCTs. Racette et al,[35] although using randomisation, only included two worksites, with quite different employee characteristics which may have introduced other sources of bias.[36] Sorensel et al[36 39–41] collected more dietary data than was presented in the report, without clear justification for why specific outcomes were reported. Nearly all of the included randomised studies did not provide details on the process of randomisation (selection bias). Two studies[37 42 45] were at low risk of performance bias where blinding was not possible but unlikely to affect objective outcomes.

In the non-randomised studies, high risk of bias in more than one domain was seen in 4[26 34 38 43 44] of the 12 included studies. One study[29] had errors in the numerical data presented which was cause for concern (number of entrees sold and mean number of entrees sold/day). Only two studies were judged at low risk of bias across all domains.[25 31] Half of the studies were at low risk of bias for confounding variables.

## DISCUSSION
### Statement of principal findings

Choice architecture interventions involving a proximity element appear to be frequently effective at changing behaviour. This was found in some studies that used proximity interventions alone or as part of choice architecture or complex interventions. Interventions including an availability element were generally reported to be successful at changing behaviour. However, none of the studies examined availability alone. A sizing intervention was effective in the only study identified examining this type of intervention.

There was no strong evidence for the effect of pricing on behavioural change in our review. No additional effect of pricing in a sizing and pricing intervention compared with sizing alone, and one complex intervention including pricing which was reported to be effective, but had a high risk of bias. Labelling alone was generally not effective at changing behaviour, although it was a popular addition to other interventions of which most were successful.

### Strengths and weaknesses of the study

Because of the relatively small number of papers that we found during a scoping search, we made the decision to broaden the inclusion criteria in two ways. First, we have included studies in which complex interventions with non-choice architecture elements were implemented and it is not always possible to draw conclusions about the contribution of the choice architecture elements to the outcomes. Second, we have included studies in which not all the workplaces were healthcare settings. We did this because this gives us some evidence of interventions which are feasible in healthcare settings; however, because data were not disaggregated, we cannot be sure that outcomes were similar across the different sorts of workplaces included (ie, an element that was effective overall, may have been less effective or not effective in the healthcare settings; or an element that was not effective overall may have been effective in the healthcare settings). We conducted a comprehensive search across major electronic databases relevant to the topic of this review. Each included study in this review was comprehensively selected, assessed, data extracted and quality assessed by two review authors independently to minimise potential biases in the review process. Where data of relevance were missing, either to allow assessment of eligibility or at the data extraction stage and assessment of bias, the review authors (OO, LA, OU) contacted the study authors for further information.

### Strengths and weaknesses in relation to other studies, discussing important differences in results

When Hollands et al[20] appraised the choice architecture literature in mid-2011, for the purposes of developing the typology we have used in this review, they noted that 40% of all study reports examined interventions which involved point-of-choice labelling and/or prompting elements.[20] We found labelling and proximity elements were the most popular type of choice architecture employed in our included studies, both alone or in combination with other intervention elements. The difference might be due to the scope of the literature examined, we were restricted to those aiming to change dietary behaviour while Hollands et al[19] looked more broadly at a range of behaviours.

Our study is specifically focused on employees of healthcare settings, a unique population who are likely to be more health conscious than the general population. For this reason, our findings are not necessarily generalisable outside these settings and others' findings may not be relevant for healthcare staff.

The finding of this review found mixed results for labelling interventions, suggesting that they are unlikely to be effective in isolation in these settings. A recent systematic review evaluating the effect of restaurant menu calorie labelling found 19 studies. Meta-analysis of these studies demonstrated an 18 calorie reduction in meals ordered when all studies were combined. However, when just the six controlled studies were examined, labelling was no longer associated with a significant difference in calories ordered.[60] Another systematic review found that menu labelling with calorie information alone did not reduce the number of calories selected or consumed,

**Table 5** Risk of bias in randomised controlled trials

| | Selection bias | | Performance bias | Detection bias | Attrition bias | | |
|---|---|---|---|---|---|---|---|
| Study | Random sequence generation | Allocation concealment | Blinding of participants + personnel (subjective/ objective outcomes) | Blinding of outcome assessment (subjective/ objective outcomes) | Incomplete outcome data (subjective/objective outcomes) | Selective outcome reporting | Other bias |
| Beresford *et al* [46] | No details | No details | Blinding not possible | No details. Lack of blinding may affect subjective outcomes | Numbers provided but reasons not fully explained | Outcomes reported as in method. No protocol | No other bias |
| Racette *et al* [35] | No details | No details | Blinding not possible | No details. Lack of blinding may affect subjective outcomes | Numbers and reasons provided | No protocol | Randomisation was only over two sites. Demographical difference between groups |
| Sorensen *et al* [47] | No details | No details | Blinding not possible | No details | Numbers provided and gives some reason for questionnaires being eliminated from analyses | Food Frequency Questionnaire was used but only fibre and fat reported | No other bias |
| Sorensen [36] | No details | No details | Blinding not possible | No details | Not reported but analysis presented for all sample (not restricted for completers) | 'Additional dietary data, not presented here, were collected by means of the Food Frequency Questionnaire' | No significant differences among the groups at baseline. All analyses were computed by taking into consideration the nesting of employees in worksites |
| Stites [37] | Using a computer program | No details | Blinding not possible | No details | Mainly reported and reasons explained, for one participant, no reason explained | Outcomes reported as in method. No protocol | No other bias |
| Vermeer [42] | No details | No details | Blinding not possible | No details | Numbers provided but reasons not fully explained | Outcomes reported as in method. No protocol | No other bias |

Green, low risk of bias; orange, unclear risk of bias; red, high risk of bias; '*quote from publication*'.

**Table 6** Risk of bias is non-randomised studies

| | Selection bias | Confounding variables | Performance bias | Detection bias | Attrition bias | Selective outcome |
|---|---|---|---|---|---|---|
| Study | Selection of participants | Confounding variables | Measurement of exposure | Blinding of outcome assessment (subjective/ objective outcomes) | Incomplete outcome data (subjective/ objective outcomes) | Selective outcome reporting |
| Dorresteijn | Data were collected prospectively. | Confounding variables (weather) were considered. | All were exposed, performance bias is not affected by exposure. | Blinding not possible and unlikely to affect objective outcomes. | Characteristics of study before and after the intervention are available. | All of the expected outcomes were included in the study descriptions. |
| Geaney | *'A random sample of 100 individuals took part in the study (fifty staff from each hospital).' 'n the intervention hospital, all individuals who were asked to participate agreed. Less than five in the non-intervention hospital refused to participate in the study.'* | Confounding variables (age and gender) were considered. | All were exposed, performance bias is not affected by exposure. | No details. | Cross-sectional there was no attrition. | All of the expected outcomes were included in the study descriptions. |
| Holdsworth [43] | *'Two premises applying for the HBA were unsuccessful in receiving it, and were used as a comparison group for the study.'* | Confounding variables (adjusted for age, gender, ethnicity, social class and body mass index) were considered. | All were exposed, performance bias is not affected by exposure. | Lack of blinding may affect subjective outcomes. | Numbers provided (low response rate). | All of the expected outcomes were included in the study descriptions. |
| Kwak et al [48] | *'A sample of 128 worksites meeting the inclusion criteria were selected through the Chamber of Commerce and invited by letter and telephone to take part in this study .... 12 worksites agreed to participate.'* | Confounding variables (matched based on the social economic status of their employees) were considered. | All were exposed, performance bias is not affected by exposure. | Lack of blinding may affect subjective outcomes. | *'At baseline, the average response rate to the questionnaire was 88.1% (487 of 553). At the first follow-up, the response rate average was 82.5% (n=376) and at the second follow- up 75.8% (n=303). During the course of the project, one of the worksites in the intervention group underwent reorganization; half of the employees were made redundant, thereby causing most of the dropout in our study. There was no selective dropout at the first follow-up. At the second follow-up, smokers were more likely to discontinue the study than non-smokers.'* | More items collected than reported. |

Continued

**Table 6** Continued

| Study | Selection bias | | Performance bias | Detection bias | Attrition bias | Selective outcome reporting |
|---|---|---|---|---|---|---|
| | Selection of participants | Confounding variables | Measurement of exposure | Blinding of outcome assessment (subjective/objective outcomes) | Incomplete outcome data (subjective/objective outcomes) | |
| Lassen et al | 'The employees were randomly approached after having bought their meal and asked to participate in the study focusing on canteen food intake.' | Confounding variables (demographics, seasonal change) were not considered. | All were exposed, performance bias is not affected by exposure. | Lack of blinding unlikely to affect objective outcomes. | Evidence significant differential in number of healthcare personnel between baseline and follow-up | 'For the outcomes of refined sugar, wholegrain and plate waste a large number of participants had zero intake. These were analysed in two steps. First the probability of any intake yes/no was analysed .... then a regression model was fitted as described above with regard to the participants with a positive intake.' |
| Lowe et al [34] | 'Participants were recruited through letters distributed to cafeteria patrons that provided an overview of the study along with inclusion and exclusion criteria.' | No details. | All were exposed, performance bias is not affected by exposure. | Lack of blinding is unlikely to affect subjective outcomes. For subjective outcomes 'Dietary recalls were conducted by the Diet Assessment Center at The Pennsylvania State University.' | Numbers provided (high attrition rate). 'Data from 'Participants who did not scan their cards an average of at least four times per month were excluded from analyses.' | Reporting of data was not always as per Intervention group with no justification. |
| Macdonald et al [38] | Five centres were chosen to participate including a range of contexts (metropolitan, rural, regional). | Confounding variables (changes at the sites) were not considered. | All were exposed, performance bias is not affected by exposure. | 'Catering receipts received included generic food types such as sandwiches. Categorisation into food types in these instances were based on inferences of the project dietitian.' 'The number of serves provided where invoices listed general categories; for example, "large fruit platter" were made from inferences based on cost by the project dietitian.' | 'Two ACCHOs were able to provide catering receipts at baseline and follow-up.' | All of the expected outcomes were included in the study descriptions. |
| Meyers [27] | Data were collected prospectively. | No details. | All were exposed, performance bias is not affected by exposure. | 'Observations were carried out by undergraduate research assistants.' | N/A. | All of the expected outcomes were included in the study descriptions. |
| Mlich [28] | 'From original total pool of sampled employees 50 subjects were randomly sampled for each body size in each experimental condition.' | Confounding variables (weather, obesity and price change) were considered. | All were exposed, performance bias is not affected by exposure. | Outcome assessed by assessing food items on plate then calculating caloric value. | N/A. | All of the expected outcomes were included in the study descriptions. |
| Sato et al [29] | Data were prospectively collected. | Confounding variables (price change) were not considered. | All were exposed, performance bias is not affected by exposure. | Lack of blinding is unlikely to affect objective outcomes. | N/A. | All of the expected outcomes were included in the study descriptions. |

Continued

**Table 6** Continued

| Study | Selection bias — Selection of participants | Confounding variables | Performance bias — Measurement of exposure | Detection bias — Blinding of outcome assessment (subjective/objective outcomes) | Attrition bias — Incomplete outcome data (subjective/objective outcomes) | Selective outcome reporting |
|---|---|---|---|---|---|---|
| Thorndike et al [30] | *'Employee platinum plate users who made a purchase in the cafeteria at least three times during each 3 month period from baseline to the end of follow-up were included in the study cohort.'* | Confounding variables (age, gender, race, job type, and work status) were considered. | All were exposed, performance bias is not affected by exposure. | Lack of blinding is unlikely to affect objective outcomes. | *'1.7% of employees missing any sociodemographic data were excluded from the analysis.'* Information about the number of participants before and after the study exists, however the cohort was not identical. | All of the expected outcomes were included in the study descriptions. |
| Van Kleef [31] | *'Carried out in hospital staff restaurant and visitors are allowed to purchase displayed snacks at a self-service checkout counter throughout the day over a fourweek period. Data were collected prospectively.'* | No details. | All were exposed, performance bias is not affected by exposure. | Lack of blinding is unlikely to affect objective outcomes. | N/A. | All of the expected outcomes were included in the study descriptions. |

Green, low risk of bias; grey, not applicable; N/A: not applicable; orange, unclear risk of bias; red, high risk of bias; *'quote from publication'*.

although the addition of information to assist consumers in applying nutritional information was effective.[61] In a systematic review of RCTs that included food labels and excluded menu labelling, nine studies were identified and meta-analysis concluded that food labelling was effective at changing dietary behaviour although not at reducing calorie consumption.[62] Menu energy labelling across socioeconomic groups was evaluated in a recent review.[63] The findings suggested that any positive benefit of this kind of labelling may only apply to higher socioeconomic groups.[63] The findings of this review in the context of previous literature suggest that labelling alone should not be recommended for implementation to improve healthy food and drink purchasing of healthcare staff.

A large systematic review evaluated supermarket and grocery store interventions to increase consumption of healthy food and/or beverages. Analysis of 42 studies found that consumers responded to economic incentives including discounts, vouchers and subsidies.[64] Similarly, a systematic review of interventions at point-of-sale to encourage healthier food purchasing found evidence for monetary incentives as effective at changing behaviour, but that there was no strong evidence for the other intervention types they examined.[65] We found mixed evidence on pricing in healthcare settings in this review.

A systematic review of interventions carried out in restaurant settings to promote healthy eating stated that the most effective interventions were point-of-purchase information with increased availability of healthy choices.[66] We also found that increasing availability of healthy options alongside other intervention elements was a successful strategy for driving behaviour change in healthcare settings.

### Meaning of the study: possible explanations and implications for clinicians and policymakers

On the basis of this systematic review, we were able to recommend some general principles that are likely to be effective when designing choice architecture interventions for NHS organisations to PHE. These are 'proximity' interventions in which behavioural options are made easier or harder to engage with, requiring reduced or increased effort and 'availability' interventions in which healthy behavioural options are added within a given microenvironment. In addition, we are able to suggest 'sizing' interventions in which the size or quantity of the product on offer is changed based on one effective study with low risk of bias. Finally, we can recommend specific examples of choice architecture interventions that have been effective in studies with low risk of bias. These recommendations are made in a companion format (please see supplementary material 3), designed for the audience who make decisions about catering in NHS organisations.

### Other outcomes relevant to decision-makers

We examined primary outcomes relating to dietary consumption and purchasing of food and drink. However, outcomes that are important to decision-makers include

customer satisfaction and profitability. One of our included studies which examined various arrangements and assortments of snacks on offer at the checkout counter in a hospital staff restaurant found in a survey of 92 participants that the more visible the healthy snacks were in terms of location (ie, on top shelves) and proportion (ie, 75% healthy vs 25% healthy) the more attractive the entire selection was perceived to be.[31] Another included study noted that the presentation of caloric values of the food significantly decreased the total number of calories bought without significantly affecting the total money spent.[28]

In a study of nine early adopters of the Hospital Healthier Food Initiative and the Health and Sustainability Guidelines in the USA, the main barriers to implementation were reported to be customer complaints, a shortage of food and drinks that met the requirements of the Initiative/Guidelines and concern over profitability. However, most respondents had been able to overcome these barriers and it was noted that customer volumes and sales were reported to have increased subsequent to implementation which offset initial investments necessary to successfully meet the requirements of the Initiative/Guidelines.[16]

A study in which a 'healthful food station' was introduced to a worksite cafeteria in an academic medical centre in the USA found that some customers reported that they used the cafeteria more often due to the presence of the station. Others had only started using the cafeteria because the station had been introduced. This study also found that the healthful food station generated gross profit.[56]

### Unanswered questions and future research
Reporting of studies was generally poor. For instance, both randomised and non-randomised studies did not generally report on domains that may lead to selection bias. We are generally unsure whether subjective outcomes were collected by blinded assessors which may lead to detection bias. Overall, the methodological quality of included studies was low and future research should clearly report on this. Therefore, the results of this review should be interpreted with caution.

In healthcare settings, increasing availability of healthy options alongside other intervention elements was found to be successful in improving behaviour (specifically increasing consumption of fruit and vegetables). However, we do not know the effect of increasing availability alone.

We could not come to a clear conclusion whether a pricing element is clearly successful in improving healthy food and drink purchasing and consumption in healthcare settings.

No studies investigated the effect of ambience or priming elements on modifying dietary behaviour in healthcare settings.

**Acknowledgements** The authors thank Aileen Clarke, Warwick Medical School; Ruth Breese, South Warwickshire NHS Foundation Trust; Elizabeth Atherton, Food for Life Programme, Soil Association; Laura Brown, Sarah Golding and Tim Chadborn, Public Health England, all of whom gave comments and feedback that have supported this work.

**Contributors** LA-K and OO designed the protocol. LA-K, OO and SJ designed and ran the searches. LA and OO performed the initial screening. LA-K, OO and OAU screened full-text articles. LA-K, OO, OAU and RW extracted data from included studies. LA-K and OO synthesized extracted data and wrote the first draft of the manuscript. All authors contributed to the final submitted manuscript.

**Funding** This piece of research was funded by Public Health England Behavioural Insights Team who gave feedback on the study protocol. LA and OO are supported by the National Institute of Health Research (NIHR) Collaboration for Applied Health Research and Care West Midlands initiative. RW was supported by University of Warwick through its Global Research Priority on Food. SJ was supported by the University of Warwick.

**Disclaimer** The funder had no role in the collection, analysis or interpretation of data, in the writing of the report or decision to submit the article for publication. The views expressed are those of the authors and not necessarily those of the NHS, the NIHR or the Department of Health.

**Competing interests** None declared.

**Patient consent** Not required.

**Provenance and peer review** Not commissioned; externally peer reviewed.

**Data sharing statement** No additional data available.

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
