## [Reviewer comments · BMJ Open]

ARTICLE DETAILS

TITLE (PROVISIONAL)	Choice architecture interventions to improve diet and/or dietary behaviour by Healthcare staff in high-income countries: a systematic review
AUTHORS	Al-Khudairy, Lena; Uthman, Olalekan; Walmsley, Rosemary; Johnson, Samantha; Oyeboode, Oyinlola

VERSION 1 – REVIEW

REVIEWER	Julia Allan University of Aberdeen, UK
REVIEW RETURNED	14-May-2018

GENERAL COMMENTS	Bmjopen-2018-023687 Choice architecture interventions to improve diet and/or dietary behaviour by healthcare staff in high-income countries” A systematic review This is a useful and high quality systematic review conducted in line with best practice guidelines. The topic area is an important one. The NHS is the largest employer in the UK and most of the general population pass through hospitals at some point in their lives, meaning that healthcare environments are a key setting for choice architecture interventions. A couple of very minor pedantic points; “improving behaviour” (page 12, line18) is very subjective. Replace with more concrete statement (e.g. “increasing x, y and z”) “no statistical significance” should be replaced with “statistically significant difference between x and y”. The manuscript has quite a few typos and needs a thorough proofread prior to publication; Abstract line 10, 17 Strengths and limitations line 32 Intro line 7 Patient and public involvement line 22 Multiple workplaces line 16 Discussion page 10, line 4 Discussion page 11, line 13 Overall, a valuable addition to the literature.
---

REVIEWER	Miranda Blake Research Fellow, Deakin University, Australia
REVIEW RETURNED	17-Aug-2018

GENERAL COMMENTS	Overarching This paper addresses a fairly specific but practice-relevant research question. Recommendations for health-care settings in high income countries are clear. Further explanation of inclusion/exclusion criteria and focus on the unique characteristics of the study population is warranted. Major Introduction You make it clear that the review was commissioned for NHS, but your search methods and results are transferrable to healthcare organisations in high income countries. You could therefore generalise the introduction more to appeal to international audiences. Methods p.4 Line 20- Please describe how you defined “healthy diet” interventions. Did this follow national dietary guidelines recommendations? Did you exclude food safety interventions or specialist diets, e.g. promotion of gluten-free or “paleo” foods? If interventions promoted “healthier” options was this sufficient (e.g. smaller portions of hot chips are not “healthy” but they are “healthier”)? p.4 Line 22 Please insert brief rationale for use of Hollands typology (as opposed to another, e.g. Kraak et al. A novel marketing mix and choice architecture framework to nudge restaurant customers toward healthy food environments to reduce obesity in the United States. Obes Rev. 2017). p.4 Line 30-31. If you are excluding calorie intake due to poor validity, but including other dietary measures, would be helpful to have evidence supporting the validity of these other measures, including any evidence explicitly suggesting they are more valid than calorie intake. Results Table 3: This table summarises the most important information from the paper. It would be useful if you could add another column to the right which summarises the effectiveness of each element (e.g. labelling) across all studies. You could have categories for insufficient evidence as well to highlight research priorities. The current Y/N effectiveness categorisation per study does not explicitly indicate that some interventions could worsen desired outcomes. From the rest of the results it doesn’t appear this came up, but replacing with other symbols that allow for unfavourable as well as neutral outcomes would be clearer e.g. ✓ (outcome in desired direction), X (outcome in undesired direction), - (neutral=no effect). Discussion p.10 lines 22-26. Suggest further discussion of the uniqueness of the population. It would be helpful to bring back some of the information from the introduction and further expand on the characteristics of this population (e.g. potentially more health conscious and receptive to health messages), and therefore the generalisability of these results to other settings. This does not need to be negatively framed, you could talk about the importance of identifying relevant interventions considering population, context and resources.
---

	Minor p. 2 Line 10. This is a bit confusing. Suggest change to “We expanded the choice architecture typology from Hollands et al. (2013) to include interventions that altered the price.” p.3 Line 31 Insert “-“ after “purchasing” p.3 Line 37-40. You have made the rationale for this review clear, but a formal review question (following PICOS, as you have in Methods) would be helpful for the reader. p.4 Line 6- Insert “and” before “Web of Science” p.4 Line 7- Please cite relevant systematic reviews p.5 line 22, Delete “that the” p.6 line 7- Citation for Australian study missing. p.8 Line 12- What were the key outcomes in the dietary recall data?
--	---

VERSION 1 – AUTHOR RESPONSE

Reviewer 1	
This is a useful and high quality systematic review conducted in line with best practice guidelines. The topic area is an important one. The NHS is the largest employer in the UK and most of the general population pass through hospitals at some point in their lives, meaning that healthcare environments are a key setting for choice architecture interventions. A couple of very minor pedantic points;	Thank you for your comments.
“improving behaviour” (page 12, line18) is very subjective. Replace with more concrete statement (e.g. “increasing x, y and z”)	Amended as below: In health care settings, increasing availability of healthy options alongside other intervention elements was found to be successful in improving behaviour (specifically increasing consumption of fruit and vegetables).
“no statistical significance” should be replaced with “statistically significant difference between x and y”.	Changed as suggested
The manuscript has quite a few typos and needs a thorough proofread prior to publication; Abstract line 10, 17 Strengths and limitations line 32 Intro line 7 Patient and public involvement line 22 Multiple workplaces line 16 Discussion page 10, line 4 Discussion page 11, line 13	Thank you- proof-reading completed.
Overall, a valuable addition to the literature.	Thank you!
Reviewer 2	

Overarching This paper addresses a fairly specific but practice-relevant research question. Recommendations for health-care settings in high income countries are clear. Further explanation of inclusion/ exclusion criteria and focus on the unique characteristics of the study population is warranted.	Thank you for your comments.
Introduction You make it clear that the review was commissioned for NHS, but your search methods and results are transferrable to healthcare organisations in high income countries. You could therefore generalise the introduction more to appeal to international audiences.	I have sign-posted in paragraph two of the introduction to highlight relevance to health care settings worldwide. The first two sentences of paragraph three are also generally relevant to international audiences.
Methods p.4 Line 20- Please describe how you defined “healthy diet” interventions. Did this follow national dietary guidelines recommendations? Did you exclude food safety interventions or specialist diets, e.g. promotion of gluten-free or “paleo” foods? If interventions promoted “healthier” options was this sufficient (e.g. smaller portions of hot chips are not “healthy” but they are “healthier”)?	Thanks for this. I have changed it to any diet interventions as actually we just looked for dietary interventions aiming to change purchasing or consumption and didn’t exclude anything according to a definition of what we could consider healthy. “Interventions - Behavioural interventions that aimed to change dietary purchasing and/or consumption of staff were included.”
p.4 Line 22 Please insert brief rationale for use of Hollands typology (as opposed to another, e.g. Kraak et al. A novel marketing mix and choice architecture framework to nudge restaurant customers toward healthy food environments to reduce obesity in the United States. Obes Rev. 2017).	Sentence added below: We chose this framework in discussion with PHE, as it has previously used to examine dietary interventions and was applicable to the context (health care settings).
p.4 Line 30-31. If you are excluding calorie intake due to poor validity, but including other dietary measures, would be helpful to have evidence supporting the validity of these other measures, including any evidence explicitly suggesting they are more valid than calorie intake.	We have stated that we extracted outcomes that were dietary purchasing, or dietary intake assessed by validated tools and reference 21 (given) is a book which gives the evidence for which nutritional outcomes are validated (ie: diet records and 24-hour recalls). This is in line with what Cochrane reviews reporting dietary outcomes find acceptable. Nb: it is not calorie intake which has poor validity it is *self-reported* calorie intake which has been shown to be poorly correlated with energy biomarkers. Diet records and 24-hour dietary recall have been validated against gold

standards eg: doubly labelled water/ 24-hour urine. Reference 21 covers this information but specific references for your interest:

Coulston, A.M. and Boushey, C.J. (Eds.). (2008). Nutrition in the Prevention and Treatment of Disease, Academic Press.

Feskanich, D., Rimm, E.B., Giovannucci, E.L., Colditz, G.A., Stampfer, M.J., Litin, L.B. and Willett, W.C. (1993). Reproducibility and validity of food intake measurements from a semiquantitative food frequency questionnaire. *Journal of the American Dietetic Association*, 93(7): 790-796.

Freedman, L.S., Kipnis, V., Schatzkin, A., Tasevska, N. and Potischman, N. (2010). Can we use biomarkers in combination with self-reports to strengthen the analysis of nutritional epidemiologic studies? *Epidemiologic Perspectives & Innovations*, 7(1): 2.

Martin-Moreno, J.M., Boyle, P., Gorgojo, L., Maisonneuve, P., Fernandez-Rodriguez, J.C., Salvini, S. and Willett, W.C. (1993). Development and validation of a food frequency questionnaire in Spain. *International Journal of Epidemiology*, 22(3): 512- 519.

Decarli, A., Franceschi, S., Ferraroni, M., Gnagnarella, P., Parpinel, M.T., Vecchia, C.L., Negri, E., Salvini, S., Falcini, F. and Giacosa, A. (1996). Validation of a food frequency questionnaire to assess dietary intakes in cancer studies in Italy results for specific nutrients. *Annals of Epidemiology*, 6(2): 110-118.

Boeing, H., Bohlscheid-Thomas, S., Voss, S., Schneeweiss, S. and Wahrendorf, J. (1997). The relative validity of vitamin intakes derived from a food frequency questionnaire compared to 24-hour recalls and biological measurements: results from the EPIC pilot study in Germany. *European Prospective Investigation into Cancer and Nutrition. International journal of epidemiology*, 26(suppl 1): S82.

Kroke, A., Klipstein-Grobusch, K., Voss, S., Möseneder, J., Thielecke, F., Noack, R. and Boeing, H. (1999). Validation of a self-administered food-frequency questionnaire administered in the European Prospective Investigation into Cancer and Nutrition (EPIC) Study: comparison of energy, protein, and macronutrient intakes estimated with the doubly labeled water, urinary nitrogen, and repeated

	24-h dietary recall methods. The American Journal of Clinical Nutrition, 70(4): 439-447. Resnicow, K., Odom, E., Wang, T., Dudley, W.N., Mitchell, D., Vaughan, R., Jackson, A. and Baranowski, T. (2000). Validation of three food frequency questionnaires and 24-hour recalls with serum carotenoid levels in a sample of African-American adults. American Journal of Epidemiology, 152(11): 1072-1080. Willett, W.C. and Stampfer, M.J. (1986). Total energy intake: implications for epidemiological analysis. American Journal of Epidemiology, 124: 17-27. Schatzkin, A., Kipnis, V., Carroll, R.J., Midthune, D., Subar, A.F., Bingham, S., Schoeller, D.A., Troiano, R.P. and Freedman, L.S. (2003). A comparison of a food frequency questionnaire with a 24-hour recall for use in an epidemiological cohort study: results from the biomarker-based Observing Protein and Energy Nutrition (OPEN) study. International Journal of Epidemiology, 32(6): 1054-1062.
Results Table 3: This table summarises the most important information from the paper. It would be useful if you could add another column to the right which summarises the effectiveness of each element (e.g. labelling) across all studies. You could have categories for insufficient evidence as well to highlight research priorities. The current Y/N effectiveness categorisation per study does not explicitly indicate that some interventions could worsen desired outcomes. From the rest of the results it doesn't appear this came up, but replacing with other symbols that allow for unfavourable as well as neutral outcomes would be clearer e.g. ✓ (outcome in desired direction), X (outcome in undesired direction), - (neutral= no effect).	Although I think these are really good suggestions which we have seriously considered, in the end, I haven't made the changes suggested for the following reasons.  1. Because we summarise the effectiveness of each element in the text using data from Tables 3 *and* Tables 4, 5 and 6 (ie: also information from the studies of complex interventions and whether or not the studies presented strong evidence or evidence was weak) so actually I think presenting it alongside Table 3 might be confusing as people would look to the corresponding columns and feel that they couldn't clearly see how the conclusions in the far right column had been reached. Better to read the text for the summary which is clearly repeated including in the abstract and "what this paper adds". 2. Because, as you say- it didn't actually come up that there were any changes in the opposite direction to what was being intended and also because (relating to your point above), in fact we looked at whether purchasing or consumption were changed by the intervention

	without having spent time defining what we thought “healthier” meant, instead going with what the study authors aims were. This means that Y/N for effect seems more accurate than giving a tick/cross which implies judgement of the behaviour. If desired (editor opinion?) I am more motivated to replace the N/Y with -/✓ than to try and summarise within table 3, which I think will end up confusing.
Discussion p.10 lines 22-26. Suggest further discussion of the uniqueness of the population. It would be helpful to bring back some of the information from the introduction and further expand on the characteristics of this population (e.g. potentially more health conscious and receptive to health messages), and therefore the generalisability of these results to other settings. This does not need to be negatively framed, you could talk about the importance of identifying relevant interventions considering population, context and resources.	A paragraph added under strengths and weaknesses to highlight this issue for the reader. “Our study is specifically focused on employees of health care settings, a unique population who are likely to be more health conscious than the general population. For this reason, our findings are not necessarily generalizable outside these settings and other’s findings may not be relevant for health care staff.”
Minor p. 2 Line 10. This is a bit confusing. Suggest change to “We expanded the choice architecture typology from Hollands et al. (2013) to include interventions that altered the price.” p.3 Line 31 Insert “-“ after “purchasing” p.3 Line 37-40. You have made the rationale for this review clear, but a formal review question (following PICOS, as you have in Methods) would be helpful for the reader. p.4 Line 6- Insert “and” before “Web of Science” p.4 Line 7- Please cite relevant systematic reviews p.5 line 22, Delete “that the” p.6 line 7- Citation for Australian study missing. p.8 Line 12- What were the key outcomes in the dietary recall data?	Thank you we have made many of the changes suggested. (have not inserted a PICO question as this is explicated in materials and methods) (not sure about your p8 line 12 comment?)
FORMATTING AMENDMENTS (if any)	
- Please embed your CORRESPONDING AUTHOR’S EMAIL ADDRESS in your main document file as shown in scholar one.	Done.
Please provide an 'Article summary' section consisting of the heading: 'Strengths and limitations of this study', and containing minimum of three (3) up to five (5) bullet points	Added.

that relate specifically to the study reported. This should be placed after the abstract.	
---	--

VERSION 2 – REVIEW

REVIEWER	Miranda Blake Global Obesity Centre, Deakin University, Australia
REVIEW RETURNED	04-Oct-2018

GENERAL COMMENTS	The authors have addressed comments well. I noted a few minor remaining typographical errors. p.4 line 25. Spell out “WHO” at first mention p.8 line 5 change to “no significance difference” p.10 line 4. Add in bracket) to end of line. p.11 line 45. Change “other’s” to “others” p.12 line 17. Insert “foods/and or beverages” after “healthy” as relevant. p.13 line 20. Change “cautious” to “caution”. p.9 lines 14-15. Please clarify what you mean by “dietary recall data”. What is the unit of measurement? e.g. is this serves of fruit and vegetables, mg sodium?
--